# Explainable Risk Prediction of Post-Stroke Adverse Mental Outcomes Using Machine Learning Techniques in a Population of 1780 Patients

**DOI:** 10.3390/s23187946

**Published:** 2023-09-17

**Authors:** Chien Wei Oei, Eddie Yin Kwee Ng, Matthew Hok Shan Ng, Ru-San Tan, Yam Meng Chan, Lai Gwen Chan, Udyavara Rajendra Acharya

**Affiliations:** 1Management Information Department, Office of Clinical Epidemiology, Analytics and kNowledge (OCEAN), Tan Tock Seng Hospital, Singapore 308433, Singapore; chienwei001@e.ntu.edu.sg; 2School of Mechanical and Aerospace Engineering, Nanyang Technological University, Singapore 639798, Singapore; 3Rehabilitation Research Institute of Singapore, Nanyang Technological University, Singapore 308232, Singapore; matthewnghs@gmail.com; 4National Heart Centre Singapore, Singapore 169609, Singapore; tanrsnhc@gmail.com; 5Duke-NUS Medical School, Singapore 169857, Singapore; 6Department of General Surgery, Vascular Surgery Service, Tan Tock Seng Hospital, Singapore 308433, Singapore; yam_meng_chan@ttsh.com.sg; 7Department of Psychiatry, Tan Tock Seng Hospital, Singapore 308433, Singapore; lai_gwen_chan@ttsh.com.sg; 8Lee Kong Chian School of Medicine, Nanyang Technological University, Singapore 308232, Singapore; 9School of Mathematics, Physics and Computing, University of Southern Queensland, Springfield, QLD 4305, Australia; rajendra.acharya@usq.edu.au

**Keywords:** automated risk prediction, post-stroke adverse mental outcome, artificial intelligence, machine learning

## Abstract

Post-stroke depression and anxiety, collectively known as post-stroke adverse mental outcome (PSAMO) are common sequelae of stroke. About 30% of stroke survivors develop depression and about 20% develop anxiety. Stroke survivors with PSAMO have poorer health outcomes with higher mortality and greater functional disability. In this study, we aimed to develop a machine learning (ML) model to predict the risk of PSAMO. We retrospectively studied 1780 patients with stroke who were divided into PSAMO vs. no PSAMO groups based on results of validated depression and anxiety questionnaires. The features collected included demographic and sociological data, quality of life scores, stroke-related information, medical and medication history, and comorbidities. Recursive feature elimination was used to select features to input in parallel to eight ML algorithms to train and test the model. Bayesian optimization was used for hyperparameter tuning. Shapley additive explanations (SHAP), an explainable AI (XAI) method, was applied to interpret the model. The best performing ML algorithm was gradient-boosted tree, which attained 74.7% binary classification accuracy. Feature importance calculated by SHAP produced a list of ranked important features that contributed to the prediction, which were consistent with findings of prior clinical studies. Some of these factors were modifiable, and potentially amenable to intervention at early stages of stroke to reduce the incidence of PSAMO.

## 1. Introduction

### 1.1. Background

Stroke is one of the main contributors to morbidity and mortality in developed countries [1]. In 2019, Singapore’s age-specific crude incidence rate of stroke was 257.6 per 100,000 population; stroke was the fourth leading cause of death and the leading cause of disability [2]. Post-stroke depression (PSD) and anxiety (PSA), collectively known as post-stroke adverse mental outcome (PSAMO),are common sequelae of stroke. About 30% of stroke survivors develop clinical symptoms of depression at some point following stroke [3,4]; and about 20% develop anxiety [4,5]. Stroke survivors with PSAMO have poorer health outcomes, including higher mortality [6,7] and greater functional disability [8,9].

Contributors to PSA include post-stroke fatigue [10,11], sleep disturbance [10], and psychosocial factors like nonmarital status, lack of social support, [11] living alone [12], family history of depression [13], and severity of stroke [13]. There is considerable overlap in the factors that contribute to PSD and PSA, e.g., left hemisphere lesions and cognitive impairment [14]. Indeed, patients may experience symptoms of anxiety after PSD, and vice versa [15,16,17]. Early intervention and treatment can play important roles in managing PSD and PSA. Studies have shown that antidepressants and psychotherapy in the early stages of stroke can be helpful to manage PSAMO [12,18,19]. 

### 1.2. Literature Review

Wang et al. also studied the use of ML algorithms to predict PSA, using a sample size of 395 cases and predictors such as demographics and lab results. However, risk factors that attributed to PSA were identified using conventional statistical significance that was calculated from multivariate logistic regression and not explained by the ML model. Ryu et al. and Fast et al. conducted studies in the area of PSD, and utilized a large variety of features for modeling. However, the limitation of both studies was their relatively small sample sizes (65 and 307 respectively), which impaired the generalizability of the models. Among the studies mentioned above, only Fast et al. developed a model that was explainable from which risk factors attributed to PSD could be evaluated. The other studies did not use any explainable artificial intelligence (XAI) methods.

In summary, there are limited studies that have utilized ML algorithms to develop a predictive model. Regarding the studies that have used ML algorithms, PSD and PSA have been studied separately and not as PSAMO as a whole. Most of the studies have used a small sample size and the developed model has not been explainable. Thus, we have not been able to identify the important features that the model used to arrive at such a conclusion. A summary of the studies reviewed can be found below (Table 1). 

### 1.3. Motivation and Proposed Method

We were motivated to develop a predictive model to automatically identify stroke patients at risk of PSAMO for early intervention. Compared to manual screening for PSD and PSA via questionnaire administration [20,21], which is onerous and prone to human biases, such a model could facilitate efficient screening. Machine learning (ML) have been shown to be superior to classical statistics in developing prediction models [22,23]. As such, we tested several ML algorithms, including the following: logistic regression, decision tree, gradient-boosted tree, random forest, XGBoost, CatBoost, AdaBoost, and LightGBM. Compared with recently published ML models that have either studied only PSD or PSA [24,25,26], our model attained higher accuracy for the combined PSAMO diagnosis (Table 1). 

**Table 1 sensors-23-07946-t001:** Summary of studies for automated prediction of post-stroke adverse mental outcome.

Author	Dataset	Features	Outcome	Techniques	Best Performance
Ryu et al. [24], 2022	31 PSD and 34 non-PSD cases	Medical history, demographics, neurological, cognitive, and functional test data	PSD	SVM, KNN, RF	SVM: AUC 0.711; Acc 0.70; Sens 0.742; Spec 0.517
Fast et al. [25], 2023	49 PSD and 258 non-PSD cases	Demographics, clinical, serological, and MRI data	PSD *	GBT, SVM	GBT: Balanced Acc 0.63; AUC 0.70
Wang et al. [26], 2021	395 cases	Demographics, lab results, vascular risk factors	PSA	RF, DT, SVM, stochastic gradient descent, multi-layer perceptron	RF: 18.625 Euclidean distance between anxiety scores
Current study	285 PSAMO and 1495 no PSAMO cases	Demographics, stroke-related data, surgical and medical history, etc.	PSAMO *	Logistic regression, DT, GBT, RF, XGBoost, CatBoost, AdaBoost, LightGBM	GBT: AUC 0.620; Acc 0.747; F1-score 0.341

Acc, accuracy; AUC, area under the curve; DT, decision tree; GBT, gradient-boosted tree; KNN, k-nearest neighbor; RF, random forest; Sens, sensitivity; Spec, specificity; SVM, support vector machine. * Developed models were explainable.

### 1.4. Main Contributions

To the best of our knowledge, our study is the first ML model for predicting the risk of PSAMO, instead of PSA and PSD separately. Our model was trained on a 1780-subject dataset, the largest to date, which represented a broader population and enhanced the generalizability of our results. Like [26], our model incorporated explainable artificial intelligence (XAI) that was able to highlight the most discriminative features used by the model for PSAMO risk prediction, which would be useful for targeted intervention of identified patients with high-risk features.

## 2. Methods

### 2.1. Data Collection and Study Design

The study population comprised 1780 patients who had been admitted for ischemic or hemorrhagic stroke to a tertiary care hospital, each of whom had completed an anxiety or depression screening assessment within 7–37 days from the time of stroke. The time window facilitated the inclusion of patients who had been stabilized after acute stroke management, and it was consistent with the 30-day window of prediction in the literature for identifying PSAMO during stroke recovery [27,28,29,30]. Baseline characteristics, including demographics, social information (e.g., occupation, educational level, etc.), quality of life scores, stroke-related information, medical history and comorbidities, medication history, and history of psychiatric conditions and interventions, were collected that constituted potential features to be input into the model (see Appendix A). Patients who died before 37 days were excluded. The retrospective analysis of the data had been approved by the hospital ethics review board. A summarized workflow of the proposed model can be found below (Figure 1).

### 2.2. Identification of PSAMO

We used the Hospital Anxiety and Depression Scale (HADS) [31] and the Patient Health Questionnaire (PHQ) [32] to diagnose PSA, PSD, and PSAMO among our study participants. HADS is a fourteen-item questionnaire; anxiety and depression have seven items each, with each item scored from 0 to 3. PSA and PSD were diagnosed based on scores of 7 or greater on the HADS-anxiety and HADS-depression scales, respectively (82% sensitivity and 78% specificity [33]), and PSAMO was diagnosed based on scores of 10 or greater on the HADS-total scale [34]. The PHQ has two versions. PHQ-9 is a nine-item questionnaire with a score range from 0 to 27; a score of 8 or greater indicated PSD [32] (88% sensitivity and 86% specificity for major depression [35]). PHQ-2 is an abbreviated version containing only the first two items of PHQ-9 [36]; a score of 3 or greater denoted PSD (83% sensitivity and 92% specificity for major depression [36]). In the presence of the abovementioned scenarios, the stroke patient was defined as having PSAMO; and in the absence, no PSAMO.

### 2.3. Statistical Analysis

Features with continuous values were first tested for normality using the Shapiro–Wilk test. If the distribution was normal, the Student’s *t*-test was employed to test for significance between the PSAMO and no PSAMO groups. The values were reported as mean ± standard deviation. If the distribution was non-normal, the Mann–Whitney U test was employed to compare the groups and their values were reported as medians (interquartile range). Categorical features were tested for significance using a chi-square test and were reported as counts (*n*) and percentages (%). The results of the statistical analysis helped to provide a preliminary understanding of the data collected between the 2 cohorts (PSAMO and no PSAMO).

### 2.4. Data Preprocessing and Engineering

Due to the retrospective nature of our study, features with more than 25% missing data were excluded from modeling. Missing data in the remaining features were assumed to be missing at random and were imputed using multiple imputation-chained equations [37]. For training, the test data split was 70:30. The training set was used to train and validate the model using 10-fold cross-validation; the test set was set aside as unseen data to be used to evaluate model performance after model development. For the training set, continuous features were standardized, and the calculated mean and standard deviation were applied to and transformed on the test set. Categorical features were also one-hot encoded, with similar treatments on both the training and test sets. After preprocessing the dataset, 46 features were obtained that were used for modeling. The list of features is found in Appendix A.

### 2.5. Model Development

Figure 2 is the flow diagram of the steps taken for model development.

#### 2.5.1. Recursive Feature Elimination (RFE)

We used recursive feature elimination (RFE) as a feature selector to choose the most discriminative features from a given dataset for downstream modeling. To find the optimal number of the most discriminative features, RFE iteratively eliminates less important features based on their impact on the model’s performance using the same ML algorithm that is deployed to train the model later. A basic default model of each algorithm was used during RFE to potentiality reduce overfitting of the training data. Taking support vector machine (SVM) as an example, a basic default SVM model can be used to perform RFE for feature selection. Then, the data with the selected features can be further trained using the SVM algorithm with its hyperparameter tuned using Bayesian optimization as explained in Section 2.5.3 and Section 2.5.4 below.

#### 2.5.2. Synthetic Minority Oversampling Technique (SMOTE) 

The prevalence of PSAMO in our study population was mildly imbalanced at 17%. Class imbalance can exacerbate the classification bias towards the majority class, especially in ML modeling of high-dimensional problems [38]. To address the class imbalance, we applied the synthetic minority oversampling technique (SMOTE) [39] to the dataset to increase the proportion of the minority class to 50% by synthetically creating samples. The SMOTE algorithm would randomly select a minority class sample, identify its nearest few other samples using the k-nearest neighbor technique, and then synthesize samples by interpolating feature values between the selected sample and its neighbors.

#### 2.5.3. Application of Machine Learning Algorithms

We deployed eight ML algorithms to train the model. The logistic regression ML algorithm with the least absolute shrinkage and selection operator [40] is a regularization technique for mitigating overfitting that estimates the probability of the outcome belonging to either class, akin to conventional statistical logistic regression. The other seven algorithms were tree-based algorithms. The decision tree [41] iteratively splits the data into new trees depending on the values of the features, arriving eventually at the final classifications, the leaves. In a random forest classifier [42], multiple decision tree algorithms are combined to make better aggregate predictions via an ensemble learning method called bagging [43], in which new trees are trained in parallel on subsets of the training data that have been selected randomly with replacement. This reduces overfitting and enhances model generalizability. The remaining five ML algorithms employ another ensemble learning method called boosting [44], in which training is iterated sequentially, with each new tree focusing on and learning from errors made by the preceding tree. Gradient-boosted tree [45] trains on the errors of previous iterations of trees. XGBoost incorporates a regularization element in the trees to prevent overfitting [46]. AdaBoost assigns higher weights to data points that have been wrongly classified, giving them higher importance in the next iteration [47]. CatBoost constrains model complexity by growing only symmetric trees [48]. LightGBM, a more efficient boosting algorithm, uses a gradient-based one-side sampling technique to select the optimal set of data to train the next iteration of trees, reducing computation time while maintaining model performance [49].

#### 2.5.4. Bayesian Optimization and Cross-Validation

During the training of each ML algorithm, the hyperparameters were tuned using Bayesian optimization [50,51]. Bayesian optimization evaluates past iterations and leverages on the results to iteratively explore the best possible parameter space to maximize model performance. The process stops upon reaching the maximum number of iterations or when early stopping criteria have been met. Each set of hyperparameters is evaluated using repeated stratified 10-fold cross-validation with five repeats.

### 2.6. Model Evaluation

We evaluated the model using standard performance metrics: accuracy and F1-score (the geometric mean of specificity and sensitivity). In addition, we performed a receiver operating characteristic analysis, reporting the area under the curve (AUC) as an index of the degree of discrimination between the two groups. Youden’s J adjustment [52] was performed to calculate the optimal binary classification threshold value, i.e., the Youden index, which is defined as the largest distance from the AUC curve to the line of no discrimination, for each of the ML algorithms:$$Youden\;Index={Max}_{cutoff}({Sensitivity}_{cutoff}+{Specificity}_{cutoff}-1)$$

### 2.7. Model Explanation

We applied Shapley additive explanations (SHAP) [53], a model-agnostic XAI method that uses corporative game theory to quantitate the predictive value of every feature, on the model with the best-performing ML algorithm. SHAP estimates the average marginal contribution of each feature and then ranks the features based on their respective contributions to the prediction, which is a measure of feature importance. SHAP is an XAI method that offers a global explanation, which is useful in our context to understand how different features impact the model’s predictions on average and the overall behavior of a model across the entire dataset. This helps to identify key factors that attribute to PSAMO, which is one of the research objectives of this study, and intervention plans can be developed around such factors.

### 2.8. Packages Used

The statistical analyses and modeling in this study were implemented in Python 3 programming language [39,46,48,49,50,53,54,55,56,57,58,59]. The packages used and their functions are listed in Appendix B.

## 3. Results

Descriptive statistics of the study population and comparisons between the PSAMO vs. no PSAMO groups are found in Appendix A.

The training set which consists of 69 features was fitted into RFE for feature selection. During RFE, the optimal number of the most discriminative features were selected for each ML algorithm. Figure 3 plots the average accuracy of the individual algorithms against the number of features. The optimal number of discriminative features differed among the algorithms. For instance, the decision tree algorithm attained its peak mean test accuracy using 13 features. As such, the decision tree algorithm would be trained and tuned using only the 13 features selected by RFE. The hyperparameters of the decision tree algorithms were tuned using Bayesian optimization, with each set of hyperparameters evaluated using repeated stratified 10-fold cross-validation with five repeats. The best set of hyperparameters, i.e., the decision tree algorithm with the best performance, was then evaluated on the test set, which was held out and not used during the training.

Table 2 shows the performance of all ML algorithms with the best set of hyperparameters that were used and trained on. The training set is reported with a mean and 95% confidence interval of the results obtained from the repeated stratified 10-fold cross-validation with five repeats. The test set is reported after Youden’s J adjustment. 

After training all algorithms, the best-performing model for the prediction of PSAMO was the gradient-boosted tree with accuracy, F1-score, and AUC values of 0.747, 0.341, and 0.620, respectively; the latter after adjustment of the cut-off value using the Youden Index (Table 2). 

### SHAP Explanation of the Gradient-Boosted Tree Model

SHAP was applied to the gradient-boosted tree model. Figure 4 depicts the top ten most important features calculated using the SHAP algorithm. 

## 4. Discussion

### 4.1. Model Performance

The developed ML model enabled automated classification of PSAMO vs. no PSAMO in stroke patients with 74.7% accuracy based on baseline features (as referenced against conventional manually administered HADS and PHQ tests), allowing for reproducible and expeditious risk predictions. An early prediction model would enable doctors to systematically triage stroke patients at risk of PSAMO for further confirmatory assessments and to institute early intervention where applicable to prevent or ameliorate adverse mental health outcomes. Of note, our ML model could predict the risk of PSAMO, i.e., PSD and PSA collectively, and was more accurate than similar models in the literature (Table 1). Finally, even though we had labeled our dataset based on the results of manually administered PHQ and HADS, these tests were nonetheless subject to human errors and biases [20,21]. In particular, their reliance on the Likert scale for scoring exposed them to measurement errors and tendency biases that could be sensitive to different modes of administering the tests [60]. 

The best performing model in our study is the gradient-boosted tree (GBT). The advantage of using GBT is its ability to model nonlinear relationships with complex interactions between features. GBT is also known to be more robust to outliers as compared to some other algorithms like linear regression. GBT has its limitations, for example, the potential of overfitting, especially if the depth of the tree is not well chosen as a hyperparameter. This limitation is mitigated by using Bayesian optimization to identify the optimal hyperparameters. Another limitation that arises from using GBT is the lack of interpretability due to its ensemble nature as compared to linear regression, where the model can be interpreted by its calculated coefficient. This is also mitigated by using SHAP as a model explanation algorithm to identify the important features.

There are limitations to conduct a one-to-one comparison against previous studies as our study is the first to study PSAMO, while previous studies have examined PSD or PSA separately [24,25,26]. Other studies have also utilized different evaluation metrics and ML algorithms that were not used in this study [24,26]. The closest study that could be referenced is Fast et al.’s [25] study. Fast et al.’s best performing model is also GBT. Both studies showed similar AUROC performances (0.7 ± 0.1 vs. 0.620). However, our study has a greater sample size (1780 vs. 307), which is more representative of the population, and therefore it improves the generalizability of our findings. The features that were used were also slightly different, with only lab markers and demographics as the only common features that were used in both studies. There were also similar findings in the feature importance in both studies, which is further elaborated below. 

### 4.2. Explainable Features

The most discriminative features in the dataset as assessed by SHAP (Figure 4) were consistent with the findings of prior clinical studies [61,62,63,64,65,66,67,68,69]. While many factors like recurrent stroke, the most important feature that predicted PSAMO in our study, were non-modifiable, some risk factors might be amenable to interventions. Lower educational level and household income were risk factors for PSAMO in our study, as well as in others [61,62,63,64]. This is also coherent with Fast et al.’s study [25], which also used SHAP to interpret its GBT model and its top important features are years of education and sex. Higher education enables patients to have clearer insights into their disease, participate more actively in their own care, and better manage their emotions, all of which may reduce the risk of PSAMO [63,65]. Lower income has been found to be associated with less patient participation in after-stroke care [66], possibly due to a lack of access to appropriate care. Based on our model findings, doctors can initiate intervention plans that address socioeconomic risk factors through social aid, improved access to care services, and personalized disease education. Being single was also a risk factor for PSAMO (Figure 4), which could be attributed to lower social and emotional support [67,68,69]. Enhancing community emotional support can reduce depressive symptoms [67], and encouraging patients to stay connected with a social network can help promote positive support-seeking behavior. 

### 4.3. Study Advantages and Limitations

Our study has the following advantages and limitations.

#### 4.3.1. Advantages

To the best of our knowledge, this is the first ML model that has been designed to predict the risk of PSAMO, a composite of PSD and PSA.The model predicted the risk of PSAMO with good accuracy (i.e., 74.7%).Trained on the largest PSAMO dataset to date, our model results are less susceptible to the influence of outliers, and therefore are representative of the broader stroke population.XAI-enabled model interpretability allows doctors to develop intervention plans for important risk factors for PSAMO.

#### 4.3.2. Limitations

This is a cross-sectional study and the observed associations cannot infer causality. Future expansion of this study to longitudinal data may offer stronger insights.Deep learning methods such as neural networks can be investigated in the future, which may produce better results.

## 5. Conclusions

We have demonstrated the feasibility of using ML algorithms to predict PSAMO. Our ML model predicted PSAMO reproducibly with good accuracy and at a low cost, without the need for onerous manual diagnostic tests. Our study used a large dataset to train the model, which enhanced the generalizability of the results. Moreover, the interpretation of the model was facilitated using SHAP, which identified important risk predictors of PSAMO that were consistent with published clinical studies. This provided indirect validation of our model, which would help boost confidence among potential clinician users. The added insight into the key risk factors of PSAMO offers opportunities for early intervention. In future works, we propose to study more complex algorithms, such as neural networks or other deep learning methods, which may drive improvements in classification performance.

## Figures and Tables

**Figure 1 sensors-23-07946-f001:**
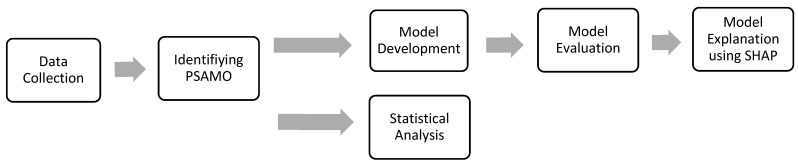
Flow diagram of the proposed model.

**Figure 2 sensors-23-07946-f002:**
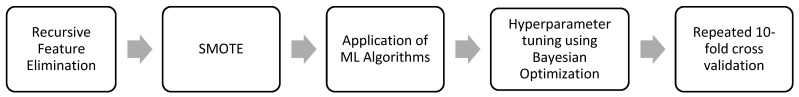
Flow diagram of model development.

**Figure 3 sensors-23-07946-f003:**
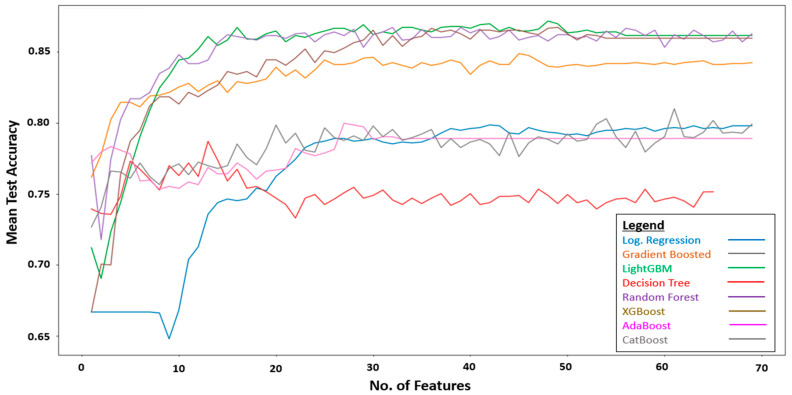
Performance of RFE using different ML algorithms.

**Figure 4 sensors-23-07946-f004:**
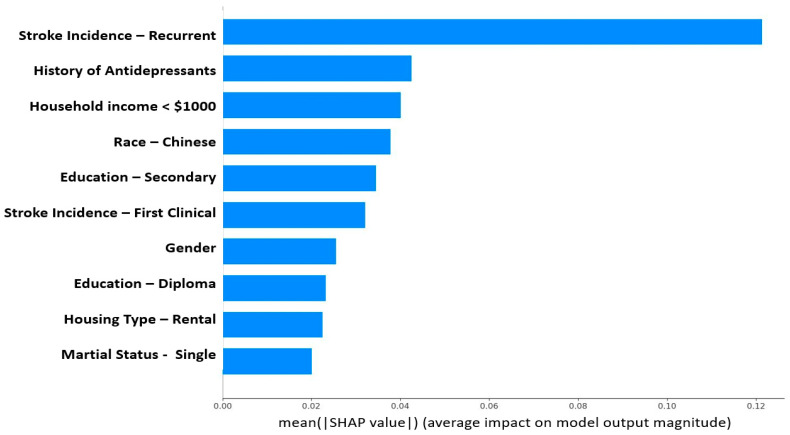
Feature importance (using SHAP values) of the developed PSAMO model. A higher SHAP value indicates that the feature has a larger contribution to the model prediction. The figure only shows the top 10 important features.

**Table 2 sensors-23-07946-t002:** Prediction performance obtained using our proposed models.

	Train Set			Test Set (after Youden’s J Adjustment)		
	Accuracy	AUROC	F1-Score	Accuracy	AUROC	F1-Score
Logistic Regression	0.696 (0.647–0.738)	0.870 (0.864–0.877)	0.658 (0.646–0.673)	0.640	0.573	0.304
Gradient-boosted tree	0.973 (0.958–0.982)	0.946 (0.932–0.957)	0.950 (0.924–0.964)	0.747	0.620	0.341
Light GBM	0.905 (0.878–0.922)	0.942 (0.924–0.964)	0.882 (0.872–0.894)	0.659	0.646	0.355
Decision tree	0.729 (0.660–0.760)	0.831 (0.810–0.844)	0.705 (0.656–0.729)	0.713	0.622	0.338
Random forest	0.800 (0.779–0.820)	0.911 (0.906–0.915)	0.779 (0.770–0.787)	0.506	0.671	0.359
XGBoost	0.899 (0.874–0.920)	0.956 (0.934–0.968)	0.877 (0.868–0.889)	0.539	0.641	0.346
AdaBoost	0.843 (0.825–0.860)	0.946 (0.939–0.948)	0.828 (0.811–0.841)	0.727	0.634	0.354
CatBoost	0.966 (0.950–0.980)	0.966 (0.952–0.970)	0.957 (0.945–0.971)	0.631	0.592	0.318

## Data Availability

Data are not available due to privacy restrictions.

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
