# Peer review of "Explainable Risk Prediction of Post-Stroke Adverse Mental Outcomes Using Machine Learning Techniques in a Population of 1780 Patients"

_sensors, 2023, doi:10.3390/s23187946_

Round 1

Reviewer 1 Report

The review file is attached. 

Author Response

Dear Reviewer, 

Thankful for taking your time to review my manuscript and provide the relevant comments for improvement.

I have uploaded my responses to your comments in the file attached.

Thank you!

Warmest Regards,
Chien Wei

Reviewer 2 Report

I believe the paper's topic is relevant, but the introduction  should  be enhanced. A literature review should be included in the introduction or as a distinct section.  Indeed, 69 papers were cited in the "References" section, and their relevance to the study should be further clarified, which helps to highlight the relevance and additional value of the proposed research. 

The presentation of the  results require further improvement and some details should be added. As example, the authors did not mention the number of features considered for the  results shown in table 2. Is the number of features same for all ML algorithms? if yes, how many features are  considered? So the experimental conditions that helped to obtained the results reported in table 2, should very detailed and justified. 

The cross validation results should be presented.

It is not clear how the results in figure 3 relate to those in table 2 or how they are discussed. Otherwise, it appears that the study is unrelated to these outcomes. 

The SHAP values for several features are displayed in Figure 4. But why were only ten features taken into account, I wonder? How low a SHAP value must the feature be before it loses all meaning? Figure 4's results appeared to be inconsistent with the tests conducted to choose the ML method.  

I believe that the comparison with state of the art should be examined in greater depth in the discussion subsection. 

The overall research design needs to be enhanced.  In particular,   sections on introduction and results need to be improved. 

Author Response

Dear Reviewer, 

Thanks for spending time to review the manuscript and to provide your insightful feedbacks.

Apologies for the slight delay as I was national service duty and have limited access to internet.

I have provided my responses as attached. Thank you!

Warmest Regards,
Chien Wei

Round 2

Reviewer 2 Report

The new version is improved compared to the previous one.